# The Effects of His Bundle Pacing Compared to Classic Resynchronization Therapy in Patients with Pacing-Induced Cardiomyopathy

**DOI:** 10.3390/jcm11195723

**Published:** 2022-09-27

**Authors:** Rafal Gardas, Krzysztof S. Golba, Tomasz Soral, Jolanta Biernat, Piotr Kulesza, Mateusz Sajdok, Kamil Zub

**Affiliations:** 1Department of Electrocardiology, Leszek Giec Upper-Silesian Medical Centre, Silesian Medical University, 40-055 Katowice, Poland; 2Department of Electrocardiology and Heart Failure, Medical University of Silesia, 40-007 Katowice, Poland

**Keywords:** conduction system pacing, His bundle pacing, pacing-induced cardiomyopathy, cardiac resynchronization therapy

## Abstract

Pacing-induced cardiomyopathy (PICM) is among the most common right ventricular pacing complications. Upgrading to cardiac resynchronization therapy (CRT) is the recommended treatment option. Conduction system pacing with His bundle pacing (HBP) has the potential to restore synchronous ventricular activation and can be an alternative to biventricular pacing (BVP). Patients with PICM scheduled for a system upgrade to CRT were included in the prospective cohort study. Either HBP or BVP was used for CRT. Electrocardiographic, clinical, and echocardiographic measurements were recorded at baseline and six-month follow-up. HBP was successful in 44 of 53 patients (83%). Thirty-nine patients with HBP and 22 with BVP completed a 6-month follow-up. HBP led to a higher reduction in QRS duration than BVP, 118.3 ± 14.20 ms vs. 150.5 ± 18.64 ms, *p* < 0.0001. The improvement in New York Heart Association (NYHA) class by one or two was more common in patients with HBP than those with BiV (*p* = 0.04). Left ventricular ejection fraction (LVEF) improved in BVP patients from 32.9 ± 7.93% to 43.9 ± 8.07%, *p* < 0.0001, and in HBP patients from 34.9 ± 6.45% to 48.6 ± 7.73%, *p* < 0.0001. The improvement in LVEF was more considerable in HBP patients than in BVP patients, *p* = 0.019. The improvement in clinical outcomes and left ventricle reverse remodeling was more significant with HBP than BVP. HBP can be a valid alternative to BVP for upgrade procedures in PICM patients.

## 1. Introduction

Right ventricular pacing (RVP) remains the primary method of permanent pacing in patients with bradycardia and preserved left ventricle (LV) function. It is simple to apply, ensures low and stable pacing parameters, and is related to a low complication rate. However, RVP leads to an abnormal and asynchronous myocardial activation, possible LV function deterioration, and the development of pacing-induced cardiomyopathy (PICM) in up to 20% of paced patients [1].

Cardiac resynchronization therapy (CRT) with biventricular pacing (BVP) can reverse PICM [2]. In the last few years, His bundle pacing (HBP) with direct conduction system pacing has been an alternative to BVP in patients with indications of CRT [3]. 

The present study compares the effects of HBP and BVP on cardiac function in patients with PICM.

## 2. Materials and Methods

### 2.1. Patient Selection

The study included a prospective, observational analysis of patients hospitalized in the Department of Electrocardiology at Professor Leszek Giec Upper-Silesian Medical Centre of the Medical University of Silesia in Katowice, Poland, with chronic RVP and heart failure symptoms considered to be caused by PICM. PICM was defined as worsening congestive heart failure accompanied by a decline in the left ventricular ejection fraction (LVEF) below 50% with an RVP burden of equal to or over 40% [4]. Patients’ medical records were carefully checked with alternative causes of LV function decline, such as myocardial infarction, valvular heart disease, frequent (20%) premature ventricular depolarizations, and atrial arrhythmias with rapid ventricular response, and when it was the case, they were excluded. Patients scheduled for a system upgrade to CRT were included in the study. The study was conducted according to the guidelines of the Declaration of Helsinki and approved by the Local Bioethics Committee of the Medical University of Silesia (KNW/0022/KB/17/18).

### 2.2. Clinical Assessment and Follow-Up

Clinical data, 12-lead electrocardiograph (ECG), and electrical stimulation parameters were assessed at the baseline, and six-month follow-up (FU) visits. The echocardiographic evaluation was performed using two-dimensional and color Doppler echocardiography (EPIQ 7 ultrasound system, Philips, Amsterdam, The Netherlands) following guidelines [5]. The left ventricle volumes and LVEF were determined using Simpson’s biplane method. The severity of mitral and tricuspid regurgitation was graded on a three-point scale (mild = 1, moderate = 2, severe = 3) with a comprehensive assessment using the regurgitation area to the atrial area ratio and proximal isovelocity surface area method. The response rate in the improvement of LVEF after six months of FU was determined by the percent increase of LVEF in specified ranges: below 5% or none, equal to or over 5%, up to 10%, equal to or over 10%, up to 20%, and equal to or over a super-responder rate of 20%.

The New York Heart Association (NYHA) functional class was used to assess clinical outcomes. Clinical response was defined as improving the NYHA functional class by one or more classes and no heart failure hospitalization (HFH). HFH was determined as hospital admission due to worsening symptoms of heart failure and requiring intravenous diuretics or intravenous inotropic medications.

QRS duration and morphology were carefully measured during RVP and HBP or BVP. Selective HBP (sHBP) and nonselective HBP (nsHBP) capture were determined according to previously published criteria [6]. Sensed R-wave amplitude, pacing threshold, lead impedance, and pacing percentage were recorded at each visit. 

### 2.3. Implantation Procedure

The decisions to use HBP or BVP and an implantable cardioverter-defibrillator (ICD) in patients with LVEF equal to or below 35% were made after careful evaluation by a heart team consisting of an electrophysiologist, a cardiologist with high expertise in echocardiography, and a heart failure specialist, as needed. The final decision to use HBP, or BVP was left at the operator’s discretion and was largely dependent on the operator’s experience with HBP. For HBP implants, the SelectSecure pacing lead (model 3830, Medtronic Inc., Minneapolis, MN, USA) was used for mapping and pacing in all cases, as previously described [7]. Predominantly, a fixed-shaped (C315HIS, Medtronic Inc., Minneapolis, MN, USA) or very rarely deflectable (C304, Medtronic) catheter was used to deliver the lead. His bundle potentials were recorded in a unipolar fashion with a Medtronic pacing system analyzer (model 2290) or an electrophysiological recording system (WorkMate Claris, Abbott, Sylmar, CA, USA). Pace mapping was used to locate the target destination when the His bundle (HB) electrogram was not recordable. The HBP lead was connected to the LV port of CRT devices or the right ventricle port. In most patients with permanent atrial fibrillation, the HBP lead was connected to an atrial port of a dual-chamber pacemaker or ICD. BVP implants were performed using a routine coronary sinus LV lead implant procedure [8]. The LV lead type (bipolar or quadripolar) and manufacturer (Abbott, Medtronic) was left to the preference of the implanting physician. All BVP devices were programmed with one-dipole LV pacing. The capture threshold <3.5 V @ 1.0 ms was accepted for HBP and BVP implant procedures. In patients with LVEF ≤35%, the decision to implant a cardioverter-defibrillator was made after careful evaluation by the treating cardiologist and implanting physician in agreement with the patient. 

### 2.4. Statistical Analysis

Continuous variables were expressed as means ± SD or median (IQR). The Kolmogorov-Smirnov test was used to determine whether continuous variables followed a normal distribution. The independent two-sample *t*-test was used to compare data between groups with correction for unequal variance (Welch test) if needed, and the paired t-test to compare data within the same group. The Wilcoxon signed-rank test was used to compare data between groups for nonparametric data. The two-way analysis of variance with the Holm-Sidak test for post hoc comparisons was used to test the differences between the means of subgroups of variables with two qualitative values: before vs. post-upgrade and HBP vs. BVP. Categorical data were presented as numbers and percentages and compared using the ꭕ2 test and ꭕ2 test for trend (the Cochran-Armitage test). Multivariate stepwise linear regression was adjusted for age and gender and was preceded by univariate linear regression to determine the independent contributions of variables to the improvement of postprocedural LVEF in six months of FU. Statistical tests were 2-sided, and a *p*-value below 0.05 was considered statistically significant.

Analyses were performed using MedCalc Statistical Software version 20.112 (MedCalc Software Ltd., Ostend, Belgium; https://www.medcalc.org, accessed on 28 August 2021) and SigmaStat 4.0 (SYSTAT Software Inc., San Jose, CA, USA) by a university lecturer in medical statistics (KSG).

## 3. Results

### 3.1. Baseline Characteristics

Between October 2016 and October 2020, we performed 3644 de novo implantations, including 683 conduction system pacing implantations, and 228 system upgrades for various reasons. In 82 consecutive patients with PICM, the upgrade procedure to CRT was performed. Fifty-three patients were scheduled for HBP implant and 29 for BVP implant. HBP was successful in 44 (83.0%) patients. Nine patients (13.3%) with unsuccessful HBP were implanted with a coronary sinus LV lead. Twenty patients were lost to follow-up, and complete clinical and echocardiography data were obtained after 6 months of the FU in 61 patients, including 39 patients with HBP and 22 with BVP.

The mean age was 72.5 ± 10.00 years, and 21.3% of the patients were females. The mean time of prior RVP was 101.5 ± 86.45 (39.0–120.0) months. The mean RVP QRS duration was 182.3 ± 15.64 ms. The mean LVEF at baseline was 34.2 ± 7.03%. The pre-RVP mean LVEF was 51.6 ± 5.72% (available for 45 patients). There were no significant differences between both groups, with a trend toward a higher incidence of atrial fibrillation and a higher NYHA functional class in HBP patients. Baseline characteristics of the study population are listed in Table 1.

### 3.2. Pacing and Procedural Outcomes

Pacing and procedural outcomes are listed in Table 2. Compared to BVP, HBP implant procedure and fluoroscopy times were significantly shorter. The HBP implant was unsuccessful in 9 (13.3%) patients: In 8 patients, the His bundle was not mapped, or only myocardial capture was achieved, and in 1, HBP was rejected due to a long HV interval and the inability to program the CRT-D device. In almost 40% of HBP patients, the conventional pacemaker was implanted. Devices with defibrillation therapy were implanted in 11 patients with HBP and 9 with BVP (*p* = 0.3).

Pacing thresholds were not significantly different between HBP and BVP at implant and during FU. A rise in the pacing threshold ≥1 V was observed in 3 patients with HBP and none with BVP. At the 6-month FU, a pacing threshold ≥2.5 V was observed in 6 patients with HBP and 1 with BVP (*p* = 0.2). 

In 1 patient with HBP, the device was extracted due to pocket infection and reimplanted on the right side. In 3 patients with HBP, the device was replaced due to battery depletion after 34 ± 12.3 months. 

### 3.3. Electrocardiographic Outcomes

sHBP was observed in 15 (38.5%) and nsHBP in 24 patients with HBP. HBP and BVP were associated with significant QRS duration reduction. At 6 months, FU HBP-paced QRS durations were significantly narrower than BVP, e.g., 118.3 ± 14.20 vs. 150.5 ± 18.64, respectively, *p* < 0.001, as shown in Figure 1.

### 3.4. Clinical and Echocardiographic Outcomes

NYHA functional status improved in patients treated with HBP, from 2.5 ± 0.60 to 1.5 ± 0.51, *p* < 0.0001, and in those treated with BVP, from 2.2 ± 0.66 to 1.7 ± 0.67, *p* = 0.002. The NYHA class improvement by 1 or 2 was more common in patients with HBP than those with BiV, where we found no improvement in 30.1% vs. 54.5%, improvement by 1 class 51.3% vs. 40.1%, and by 2 classes 17.9% vs. 4.5%, respectively, *p* = 0.0389, as shown in Figure 2A.

The new HFH was observed in 6 patients (9.8%), 3 with HBP and 3 with BVP. Regardless of HFH, 1 patient with BVP was hospitalized due to an electrical storm from recurrent ventricular tachycardia (VT), and 2 BVP patients suffered an ischemic stroke. One patient died from non-cardiac causes during follow-up.

Both upgrading approaches, HBP and BVP, significantly improved LV hemodynamics assessed with echocardiography. Compared with baseline measurements after 6 months of FU, the upgrade of the pacing mode improved LVEF in BVP patients from 32.9 ± 7.93% to 43.9 ± 8.07%, *p* < 0.0001, and in HBP patients from 34.9 ± 6.45% to 48.6 ± 7.73%, *p* < 0.0001. The LVEF improvement was more considerable in HBP patients than in BVP patients, *p* = 0.019, Figure 3. 

The response rate in LVEF improvement after six months of FU was higher in HBP than in BVP patients. The improvement was equal to or over 5%, 92.3 vs. 81.2, and equal to or over 20%, 76.9 vs. 50.0, *p* = 0.0420; for detailed data for all ranges, see Figure 2B.

There were 35 of 61 patients with LVEF equal to or lower than 35% at baseline in both HBP and BVP patients. At six months of FU, 29 (82.9%) of them achieved an LVEF over 35%. The rate of improvement was equal in both groups: *p* = 0.1488.

We observed a regression of LV volumes as an effect of both upgrade approaches. Indexed end-systolic volume reduced in HBP and BVP patients: from 66.1 ± 24.06 to 44.1 ± 16.23, *p* < 0.001, and from 64.3 ± 20.70 to 45.0 ± 13.57, *p* = 0.001, respectively. Thus, there was no difference in the reduction between the two groups. Indexed end-diastolic volume was reduced in the HBP and BVP patients, from 99.9 ± 30.61 to 84.2 ± 21.48, *p* = 0.006, and from 99.40 ± 24.4 to 79.1 ± 17.78, *p* = 0.049, respectively. Thus, there was also no difference in the reduction between the two groups.

The reduction of mitral regurgitation severity by 1+ or 2+ was more common in patients with HBP than in patients with BVP. We observed patients with no improvement (50.0% vs. 71.4%), patients with improvement by 1+ (41.2% vs. 28.6%), and patients with improvement by 2+ (8.8% vs. 0.0%), respectively (*p* = 0.0686); see Figure 2C.

Baseline data during RVP: QRS duration, LVEF, the severity of mitral regurgitation, NYHA functional class, and an HBP versus BVP upgrade approach adjusted to age and gender were used to construct a prediction model of LVEF improvement after six months of FU. The lower baseline LVEF, lower QRS duration, and HBP proved independent predictors of LVEF improvement (Table 3 and Figure 4).

The improvement is stated as the difference between LVEF at baseline and after the upgrade at six months of follow-up. HBP—His bundle pacing, BVP—biventricular pacing, LVEF—left ventricular ejection fraction, NYHA—New York Heart Association, MR—mitral regurgitation, QRS—QRS wave complex.

## 4. Discussion

This is the first publication comparing the effects of HBP and BVP in patients with PICM.

PICM is the most common RVP complication, affecting up to 20% of permanently paced patients [1]. An upgrade to CRT is the treatment of choice. As was shown in the European Society of Cardiology survey, upgrade procedures account for almost a quarter of CRT implantation procedures [9]. In a study by Khurshid et al., 69 patients with PICM LVEF improved with BVP from 29.3% to 45.2% (*p* < 0.01) after a median of 7.0 months of FU. In 85.5% of patients, LVEF improved by ≥5% [2]. However, unlike BVP, HBP with direct conduction system capture ensures the most physiological activation pattern. As Vijayaraman et al. showed, a progression of distal conduction disease in patients with atrioventricular block even after long-standing RVP is rare, and HBP can be achieved in >90% of these patients [10], and in patients with PICM, LVEF improved from 34.3 ± 9.6% to 48.2 ± 9.8% after HBP, improving LVEF by over 5% in 75% of patients. Similar improvement in LVEF was reported by Shan et al. in a small group of 11 patients with PICM [11]. In our study, the improvement in LVEF was similar to previously published data, both with HBP [10] and BVP [2]. So far, studies comparing HBP and BVP in patients with CRT indications have not shown any superiority of either pacing modalities [12,13]. However, in our study, FU LVEF was significantly higher in HBP patients at six months. We also demonstrated that improvement in LVEF, NYHA class, and mitral regurgitation were significantly more common in patients with HBP. Moreover, HBP was one of the independent predictors of LVEF improvement.

A less complicated system (i.e., a conventional pacemaker or dual-chamber ICD) was used in 43.6% of patients with HBP; in turn, in all patients with BVP, a CRT device was implanted. It seems relevant, as Kirkfeldt et al. [14] demonstrated using a nationwide registry that a CRT device is one of the most crucial risk predictors of complications, with an odds ratio of 3.3%.

Similar to other authors, we present that reverse remodeling, understood as an improvement in LVEF after an upgrade, comes quickly in most cases (within up to six months). Additionally, it should be noted that 82.9% of patients had an LVEF low enough that it qualified them for ICD therapy at baseline. Of those patients, 57.1% received ICD. It was the case in 52.4% with HBP and 64.3% with BVP. During FU, out of these patients, 90.5% with HBP and 71.4% with BVP showed improved LVEF of over 35% and consequently no longer needed ICD according to current guidelines [15]. The rate of improvement in patients with BVP was similar to the rate observed by Khurshid et al. (61.5%) [2] but was higher in patients with HBP than reported by Vijayaraman et al. (79%) [10]. During FU, we found possibly a proarrhythmic effect of BVP in only one patient with BVP and no previous ventricular arrhythmia history of recurrent VT [16]. Considering the high probability of LVEF improvement over 35% and the recently reported lack of influence of ICD therapy on all-cause mortality in patients upgraded to CRT after long-lasting RVP [4], ICD therapy should be carefully considered at the time of upgrade.

A lower baseline LVEF and a longer baseline QRS duration indicate greater severity of LV damage [17]. We have shown that the more intensive the left ventricular descent, the less beneficial the upgrade’s effect on improving LVEF in PICM patients. It suggests that patients with a high rate of ventricular pacing should be closely monitored and referred to the upgrade procedure as soon as early signs of LV damage appear. In turn, we demonstrated that using the HBP approach in such a situation predicts more remarkable improvement after an upgrade in patients with PICM.

Among HBP’s limitations are a high pacing threshold and a possible threshold rise during FU [18]. In our study, the median thresholds were not significantly different between HBP and BVP. However, in HBP patients, we found three with an increase in the threshold over 1V. 

Our study used only HBP for conduction system pacing (CSP). Recently, left bundle branch area pacing (LBBAP) has been reported as another modality of CSP. LBBAP is associated with low and stable pacing thresholds and excellent sensing parameters [19]. As Ye et al. [20] demonstrated, LBBAP is feasible for upgrade procedures in PICM patients and is associated with improved LV function and clinical status. 

Interestingly, procedure and fluoroscopy times were significantly longer with BVP, probably reflecting the more technically complex BVP implant procedure.

## 5. Study Limitations

This study is non-randomized. In some patients, the HBP lead implant was unsuccessful due to distal conduction system disease with long HV interval or with only myocardial capture, or inability to map His bundle potential. A stepwise procedure was performed in these cases, with the BVP lead implanted in the second phase. This stepwise approach can be clinically relevant, and although not used in this study, an HBP lead implant can potentially be a bail-out solution for unsuccessful BVP lead implantation. The short observation period made the comparison of clinical outcomes, particularly pacing parameters and arrhythmia burden, insufficient. Another limitation is the number of patients lost to follow-up. 

## 6. Conclusions

HBP in PICM patients is associated with more pronounced reverse remodeling and better clinical outcomes than BVP. The HBP implant procedure is less time-consuming than the BVP procedure. More data from multicenter, prospective and, possibly, randomized trials would be needed to support the findings of the study.

## Figures and Tables

**Figure 1 jcm-11-05723-f001:**
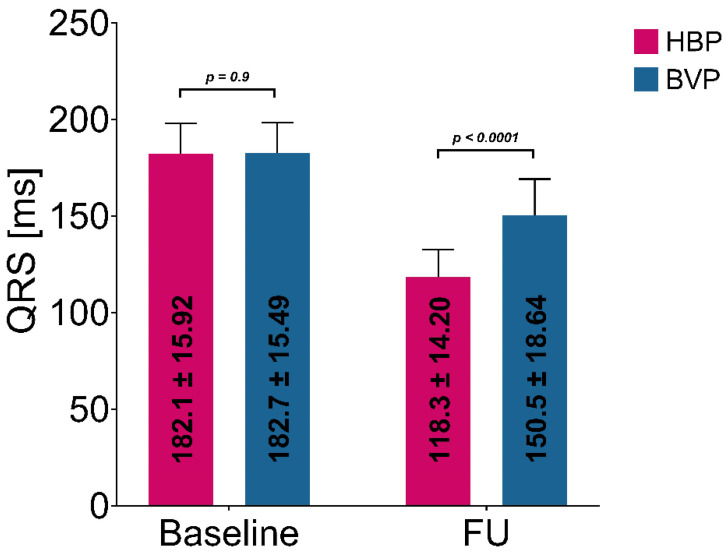
QRS duration at baseline (right ventricular paced) and follow-up (FU). HBP–His bundle pacing, BVP–biventricular pacing, QRS—QRS wave complex.

**Figure 2 jcm-11-05723-f002:**
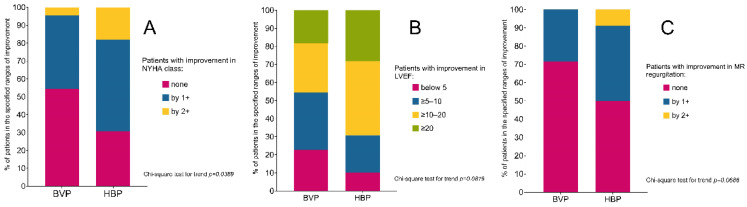
The response rates in the improvement of NYHA class (**A**), LVEF (**B**), and mitral regurgitation (**C**) after six months of FU, in specified ranges, in HBP patients compared with BVP patients. BVP—biventricular pacing, HBP—His bundle pacing, LVEF—left ventricular ejection fraction, MR–mitral regurgitation, NYHA—New York Heart Association.

**Figure 3 jcm-11-05723-f003:**
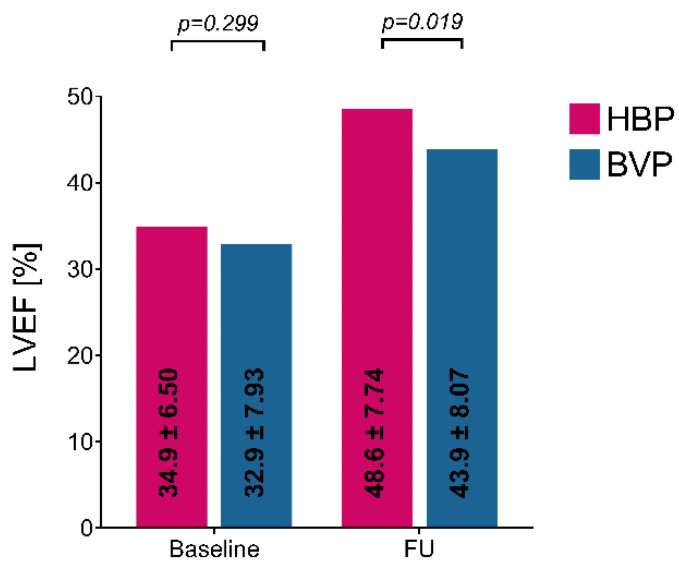
Left ventricular ejection fraction at baseline and follow-up in HBP and BVP patients. LVEF—left ventricular ejection fraction, HBP—His bundle pacing, BVP—biventricular pacing, FU—follow-up.

**Figure 4 jcm-11-05723-f004:**
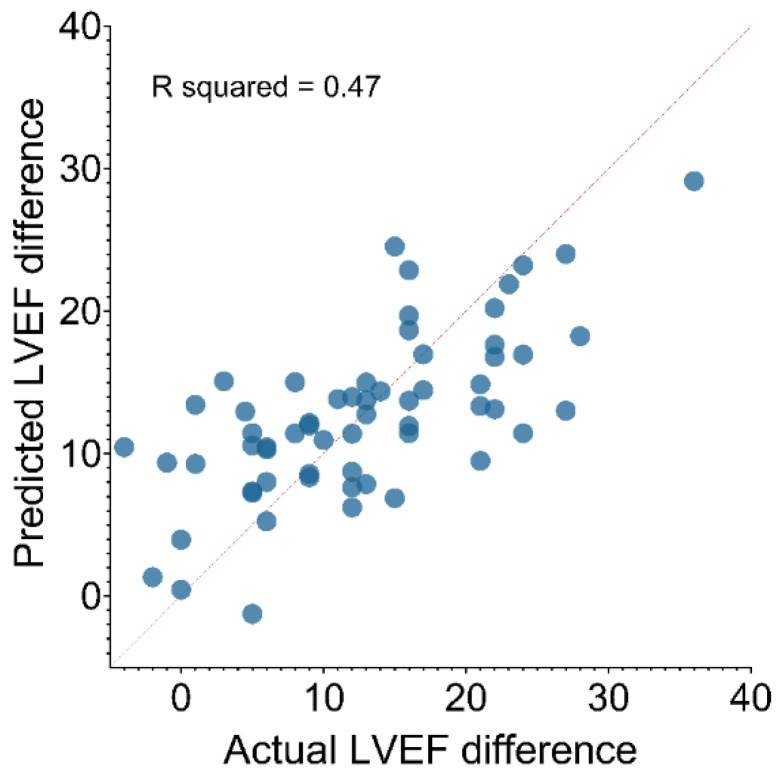
Predicted values against actual values for left ventricular ejection fraction difference before and after the upgrade procedure. LVEF—left ventricular ejection fraction.

**Table 1 jcm-11-05723-t001:** Baseline characteristics.

Characteristic	HBP (*n* = 39)	BVP (*n* = 22)	Total (*n* = 61)	*p*
Age (years)	73.8 ± 10.59	70.3 ± 8.65	72.5 ± 10.00	0.2
Females	8 (13.1%)	5 (8.2%)	13 (21.3%)	0.8
Myocardial revascularization	21 (34.4%)	9 (14.8%)	30 (49.2%)	0.6
Myocardial infarction	12 (21.3%)	6 (9.8%)	18 (29.5%)	0.8
Hypertension	30 (49.2%)	16 (26.2%)	46 (75.4%)	0.7
Diabetes	14 (22.9%)	5 (8.2%)	19 (31.1%)	0.3
Chronic kidney disease	13 (21.3%)	9 (14.8%)	22 (36.1%)	0.6
Mitral valve surgery	5 (12.8%)	1 (4.5%)	6 (9.8%)	0.3
Baseline NYHA class	2.5 ± 0.60	2.2 ± 0.66	2.4 ± 0.63	0.07
Pre-RVP LVEF (%)	51.6 ± 6.24%	51.8 ± 5.00%	51.6 ± 5.72	0.9
Baseline LVEF (%)	34.9 ± 6.45	32.9 ± 7.93	34.2 ± 7.03	0.3
≤35%	21 (34.4%)	14 (22.9%)	35 (57.4%)	0.4
36–49%	18 (29.5%)	8 (13.1%)	29 (42.6%)	0.4
Baseline LVESVi (mL)	66.1 ± 24.06	64.3 ± 20.70	65.5 ± 22.75	0.8
Baseline mitral regurgitation	1.5 ± 0.78	1.4 ± 0.68	1.4 ± 0.74	0.7
Baseline QRS duration (ms)	182.1 ± 15.92	182.7 ± 15.49	182.3 ± 15.64	0.9
% RVP pacing	90.3 ± 18.32	93.0 ± 15.97	91.2 ± 17.43	0.6
RVP time (months)	99.4. ± 83.51	105.3 ± 93.33	101.5 ± 86.45	0.8
Atrial fibrillation	25 (41.0%)	8 (13.1%)	33 (54.1%)	0.06

Values are presented as mean ± SD, median (IQR) or *n* (%). HBP—His bundle pacing, BVP—biventricular pacing, LVEF—left ventricular ejection fraction, LVESVi—indexed left ventricular end systolic volume, NYHA—New York Heart Association, RVP—right ventricular pacing, IQR—interquartile range, QRS—QRS wave complex.

**Table 2 jcm-11-05723-t002:** Procedural and pacing outcomes.

Characteristic	HBP (*n* = 39)	BVP (*n* = 22)	*p*
Procedure time [min]	79.9 ± 30.12	116.4 ± 39.07	0.0001
Fluoroscopy time [min]	11.7 ± 9.27	26.9 ± 14.65	0.0001
Pacing threshold at implant [V]	1.25 (1.0–1.775)	1.0 (1.0–1.5)	0.3
Pacing threshold at follow-up [V]	1.25 (0.75–1.5)	1.0 (0.75–1.0)	0.3
% pacing at follow-up	98.2 (95.0–100.0)	98.5 (97.0–100.0)	0.5
Device implanted			0.004
PM	15 (24.6%)	0 (0.0%)	
CRT-P	13 (21.3%)	13 (21.3%)	
CRT-D	9 (14.8%)	9 (14.8%)	
ICD	2 (3.3%)	0 (0.0%)	

Values are presented as mean ± SD, median (IQR) or *n* (%). HBP—His bundle pacing, BVP—biventricular pacing CRT-D = cardiac resynchronization therapy defibrillator, CRT-P = cardiac resynchronization therapy pacemaker, ICD = implantable cardioverter-defibrillator, min = minutes, PM = pacemaker, V = volts.

**Table 3 jcm-11-05723-t003:** Prediction of left ventricle improvement after upgrade procedure in patients with pacing-induced cardiomyopathy.

	Univariate			Multivariate		
Independent Variables	Coefficient	Std. Error	*p*	Coefficient	Std. Error	*p*
Age	0.06097	0.07639	0.4283			
Sex if male	−4.2833	1.7605	0.0184			
HBP vs. BVP upgrade approach	3.3342	1.7646	0.0643	3.9078	1.866	0.0407
LVEF % at baseline	−0.6426	0.1237	<0.0001	−0.6634	0.133	<0.0001
NYHA at baseline	−2.2583	1.5172	0.1426			
QRS duration at baseline [ms]	−0.1797	0.05449	0.0017	−0.1958	0.05691	0.0011
MR severity at baseline (0–3)	0.7208	1.0168	0.4815			

## Data Availability

The data presented in this study are available on request from the corresponding author.

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
