# Peer review of "The Effects of His Bundle Pacing Compared to Classic Resynchronization Therapy in Patients with Pacing-Induced Cardiomyopathy"

_jcm, 2022, doi:10.3390/jcm11195723_

Round 1

Reviewer 1 Report

The authors present the result of their study comparing the effect of HBP and Biv CRT on electrical resynchronization, LVEF, and clinical outcomes in patients with bradycardia and pacing-induced cardiomyopathy (PICM). They show better electrical resynchronization and a more significant increase in the LVEF and clinical improvement in the HBP group than in the BVP group. Furthermore, a multivariate analysis showed HBP, LVEF, and QRSd as predictors of LV improvement. Although the authors made a considerable effort to show the differences between HBP and Biv CRT in PICM patients, their work has substantial limitations. They are:

1/ Authors proclaim patients with alternative causes of LV function decline were excluded; however, they do not describe in detail how this was done, what were the reasons for LV decline in those excluded patients and also what were their characteristics

2/Although this was a prospective study, the authors do not show the mechanism of selecting the CRT method (HBP or Biv CRT) in PICM patients.

3/ The rate of loss to follow-up of 24% is inappropriate high for a prospective study.

4/ I am pretty surprised by the numbers shown for procedure and fluoroscopy time in the BVP group; they are too high. What is the reason for such lengthy procedure duration and fluro times? They have to be much shorter in such a cohort of patients. We have seen such long procedural and fluoro times in our experience in PICM patients in whom HBP was chosen as a primary approach, and after it failed, the BVP was done. Was this also the case here?

5/ what is the meaning of figure 4? I did not find any related explanation in the text

6/ In the discussion section, authors should focus on the discussion of results of their study, which is not the case in some parts of the discussion in the paper. 

In conclusion, despite the study being presented as a comparison of HBP and BVP in PICM patients, the lack of randomization or another mechanism of patient selection to specified groups, it is an observational study, which shows that PICM patients improve after HBP (NYHA, LVEF, LV volumes, ...) and also PICM patients with BVP improve after the procedure. Therefore, the results, as written, add only a little too current knowledge. 

Reviewer 2 Report

DEAR AUTHORS,

In my opinion, the present work is of great relevance for the future therapy of patients with pacing-induced cardiomyopathy. As described in the paper, it is a common "complication" after pacemaker implantation. I think the study is very clean and well done. Criticisms are addressed independently (not a randomized trial) and the results are discussed in detail. The relevant limitations of the different options are also discussed and mentioned. PErsönliuch I find the procedure times relatively long and am again amazed at the short battery life of His bundle pacing capable devices. Nevertheless, it is a relevant study with interesting results. Whether this work will displace CRT is still questionable, especially since the guidelines still clearly speak in favor of it and His Bundle Pacing (still) gets too little weight within the guidelines. Furthermore, the LV probe positions would have been interesting.

Summary: very good and detailed work with a clearly low number of patients, but due to the very interesting topic and the detailed elaboration of the results and the discussion a good basis for further work and comparisons.

Reviewer 3 Report

Rafal Gardas et al performed a study comparing His bundle pacing with CRT in 61 patients with pacing-induced cardiomyopathy. In this non-randomized, prospective cohort study they found HIS bundle pacing to be associated with better QRS narrowing and better LVEF improvement. This are amazing data, especially currently where there is only limited data available on CRT vs. HIS bundle pacing.

What I miss is a short description about the factors that led to the decision to perform HIS bundle pacing or CRT – operator availability? What did the heart team discuss during their meeting? Interestingly, baseline LVEF was similar between groups.

Otherwise, the manuscript is well-written with adequate methodology, statistics and the conclusions are backed up by the results. Citations are adequate.

Minor comments:

-        The authors should note in the abstracts that this is a prospective cohort study.

-        How many implantations were performed for other reasons at the observation period?

-        Incomplete follow up should be noted in the Limitations section.

-        Have the authors any data of their entire “intention to treat” analysis? (including unsuccessful HIS/CRT cases)

Round 2

Reviewer 1 Report

Dear Authors, 

there are substantial methodological shortcomings (1/selection of the patients, 2/ lack of randomization or other appropriate patient allocation to the specified group,...) that prevent the work from being presented as
a comparison study between HBP and BVP in PICM patients. 
